# Chitosan Oligosaccharide Alleviates Abnormal Glucose Metabolism without Inhibition of Hepatic Lipid Accumulation in a High-Fat Diet/Streptozotocin-Induced Diabetic Rat Model

**DOI:** 10.3390/md19070360

**Published:** 2021-06-23

**Authors:** Shing-Hwa Liu, Fan-Wen Chen, Meng-Tsan Chiang

**Affiliations:** 1Graduate Institute of Toxicology, College of Medicine, National Taiwan University, Taipei 10051, Taiwan; shinghwaliu@ntu.edu.tw; 2Department of Medical Research, China Medical University Hospital, China Medical University, Taichung 40402, Taiwan; 3Department of Pediatrics, College of Medicine, National Taiwan University Hospital, Taipei 10051, Taiwan; 4Department of Food Science, National Taiwan Ocean University, Keelung 20224, Taiwan; silverdew841256@gmail.com

**Keywords:** chitosan oligosaccharide, glucose metabolism, diabetes

## Abstract

This study investigated the effects of chitosan oligosaccharide (COS) on glucose metabolism and hepatic steatosis in a high-fat (HF) diet/streptozotocin-induced diabetic rat model. Male Wistar rats were divided into: (1) normal control (NC group), (2) HF diet (HF group), (3) streptozotocin (STZ)-induced diabetes with HF diet (DF group), and DF group supplemented with (4) 0.5% COS (D0.5F group), (5) 1% COS (D1F group), and (6) 5% COS (D5F group) for 4 weeks. COS supplementation significantly decreased the plasma glucose, BUN, creatinine, uric acid, triglyceride (TG), and total cholesterol (TC) levels, and hepatic glucose-6-phosphatase activity, and significantly increased hepatic hexokinase activity and glycogen content in diabetic rats; but the increased hepatic TG and TC levels could not be significantly decreased by COS supplementation. Supplementation of COS increased superoxide dismutase activity and decreased lipid peroxidation products in the diabetic rat livers. COS supplementation significantly increased phosphorylated AMP-activated protein kinase (AMPK) protein expression, and attenuated protein expression of hepatic phosphoenolpyruvate carboxykinase (PEPCK) and phosphorylated p38 and renal sodium-glucose cotransporter-2 (SGLT2) in diabetic rats. These results suggest that COS may possess a potential for alleviating abnormal glucose metabolism in diabetic rats through the inhibition of hepatic gluconeogenesis and lipid peroxidation and renal SGLT2 expression.

## 1. Introduction

Diabetes mellitus is a chronic and metabolic disorder. Glucose and lipid metabolism are known to be linked to each other in many ways during diabetic condition [1]. Diabetic dyslipidemia may affect glucose metabolism, leading to insulin resistance induction and β-cell function impairment [1]. Beyond drug administration, the adjustments of dietary components, such as macronutrients, micronutrients, non-nutrient phytochemicals, and additional foods, are also considered as one of the effective ways to control blood glucose that will be advantages for diabetic patients and healthy population and may limit the increasing incidence of diabetes [2].

Chitin is a natural polysaccharide polymer and widely distributed in nature, especially in marine invertebrates, insects, fungi, and yeasts. Chitosan oligosaccharides (COS) is the product of chitin or chitosan after deacetylation, the degree of polymerization (DP) of COS is less than 20 and the average molecular weight is less than 3900 Da, which was formed by the β-1,4 glycosidic linkage of glucosamine and N-acetyl glucosamine [3]. COS has low molecular weight, higher degree of deacetylation (DD), higher DP, less viscous, good water solubility, and higher absorption in the intestine, which endow it with significant biological properties and make it more applied in biomedical industries. It has been reported that COS exhibits a beneficial effect on lipid and glucose metabolism. Kim et al. have found that oral administration of COS significantly alleviates the elevated blood glucose and glycated hemoglobin (Hb1Ac) in streptozotocin (STZ)-induced diabetic rats by promoting islet β cell proliferation/neogenesis or increasing insulin secretory capacity [4]. Zheng et al. have demonstrated that treatment with COS in drinking water significantly improves hyperglycemia, insulin resistance, and lipogenesis in db/db diabetic mice [5]. COS has also been shown to inhibit rat intestinal α-glucosidase in vitro and reduce postprandial blood glucose levels in normal rats after sucrose loading test [6]. Recently, a randomized, double-blind, controlled crossover trial revealed that supplementation of COS effectively lowered postprandial blood glucose in subjects with impaired glucose tolerance and impaired fasting glucose [7]. However, the detailed effects and mechanisms of COS on glucose metabolism remain to be clarified. Moreover, the suitable doses of COS supplementation for alleviating the altered blood glucose under diabetic condition would also need to be clarified. In the present study, we investigated the effects and possible mechanisms of COS with different doses (0.5–5% in diets) as a dietary supplement on abnormal glucose metabolism in a high-fat (HF) diet/STZ-induced diabetic rat model, which induced hyperglycemia and hepatic steatosis [8,9].

## 2. Results

### 2.1. The Changes in Body Weight, Food Intake, Water Intake, Urine Volume, and Tissue Weight

As shown in Table 1, there were no significant differences in the final body weight, body weight gain, food intake, and feed efficiency among the normal control diet (NC) group and STZ diabetes + high fat diet (DF) group with or without COS (0.5–5%) (*p* > 0.05). The final body weight, body weight gain, and feed efficiency in DF group with or without COS (0.5–5%) were significantly decreased compared to high-fat diet (HF) group. Nevertheless, as compared to the NC group, the levels of water intake and urine volume were significantly increased in the DF group, but not HF group. Supplementation of COS (0.5–5%) could significantly reverse the changes in water intake and urine volume in the DF group (*p* < 0.05) (Figure 1).

The relative liver and kidney weights in the DF group were higher than that of the NC and HF groups (*p* < 0.05; Figure 1). Supplementation of COS (0.5–5%) could not reverse the changes in liver-to-body weight ratio and kidney-to-body weight ratio in the DF group (*p* > 0.05 vs. DF group; Figure 1). As compared to the NC and HF groups, the absolute and relative total, perirenal, and epididymal adipose tissue weights were significantly decreased in the DF group (*p* < 0.05; Table 2). Supplementation of COS 0.5%, but not 1 and 5%, effectively reversed the changes in total and perirenal adipose tissues in the DF group (*p* < 0.05; Table 2).

### 2.2. The Changes in Plasma Glucose, Insulin, Creatinine, Blood Urea Nitrogen (BUN), Uric Acid, Aspartate Aminotransferase (AST), Alanine Aminotransferase (ALT), and Lipids Levels

As shown in Figure 2, the levels of plasma glucose, creatinine, BUN, and uric acid in the DF group were significantly higher than that of the NC and HF groups, which could be effectively reversed by COS (0.5–5%) supplementation (*p* < 0.05 vs. DF group). The plasma insulin level in the DF group was significantly lower than that of the NC and HF groups group (*p* < 0.05; Figure 2). Supplementation of COS (0.5–5%) to the DF diabetic rats showed an increasing trend in plasma insulin level, but there was no significant difference (*p* > 0.05 vs. DF group; Figure 2).

The levels of plasma total cholesterol (TC), triglyceride (TG), and low-density lipoprotein cholesterol (LDL-C) + very low-density lipoprotein cholesterol (VLDL-C) in the DF group was significantly higher than that of the NC and HF groups, which could be effectively reversed by COS (0.5–5%) supplementation (*p* < 0.05 vs. DF group; Figure 3). The plasma high-density lipoprotein cholesterol (HDL-C) level was not changed in the DF group with or without COS (0.5–5%) supplementation (*p* > 0.05; Figure 3), but significantly decreased in the HF group (*p* < 0.05; Figure 3), compared to the NC group.

The plasma AST and ALT activities in the DF group were significantly higher than that of the NC group (*p* < 0.05; Figure 4). The plasma AST, but not ALT, activities in the HF group were also significantly higher than that of the NC group (*p* < 0.05; Figure 4). Supplementation of COS (0.5–5%) significantly reversed the change in plasma AST activity in both HF and DF groups (*p* < 0.05 vs. HF or DF group; Figure 4). However, for the analysis of plasma ALT activity, there are no significant differences among the DF, D0.5F, D1F, and D5F groups (*p* > 0.05; Figure 4). The plasma ALT activity was significantly increased in the DF diabetic rats (*p* < 0.05 vs. NC or HF group; Figure 4), which could not be significantly reversed by COS (0.5–5%) supplementation (*p* > 0.05 vs. DF group; Figure 4). Moreover, the plasma ALT activity was significantly increased in the DF diabetic rats supplemented with 5% COS compared to NC group (*p* < 0.05; Figure 4).

### 2.3. The Changes in Liver Lipids, Glycogen Content, and Glycometabolism-Related Enzymes Activities and Signaling Molecules

As shown in Table 3, the liver TG and TC levels were significantly increased in the HF and DF groups, which were slightly, but not significantly, reversed by COS (0.5–5%) supplementation.

We next examined the glycogen content and glycometabolism-related enzymes activities in the livers. The liver glycogen content (Figure 5) and hexokinase activity (Figure 6A) were not changed in both HF and DF groups. The liver glucose-6-phosphatase activity was significantly increased in DF group (Figure 6B). Supplementation of COS 1% and 5% significantly increased the liver glycogen contents (*p* < 0.05 vs. NC, HF, or DF group; Figure 5) and hexokinase activity (*p* < 0.05 vs. NC, HF, or DF group; Figure 6A), and significantly decreased the liver glucose-6-phosphatase activity in the DF diabetic rats (*p* < 0.05 vs. DF group; Figure 6B). The ratio of glucose-6-phosphatase/hexokinase was significantly increased in the DF group as compared to the NC group, which could be significantly reversed by COS (0.5–5%) supplementation (*p* < 0.05; Figure 6C). The activities of liver glucose-6-phosphate dehydrogenase were not changed in HF and DF with or without COS (0.5–5%) supplementation (*p* > 0.05 vs. NC group; Figure 6D).

Supplementation of COS 5% significantly increased the ratio of p-AMP-activated protein kinase (AMPK)α/AMPKα protein expression in the livers of the DF diabetic rats (*p* < 0.05 vs. NC, HF, or DF group; Figure 7A). The levels of protein expression of p-p38/p38 ratio, phosphoenolpyruvate carboxykinase (PEPCK), and protein kinase C (PKC)α in the livers were significantly increased in DF group, but not in HF group, which could be significantly reversed by COS (1 and 5%) supplementation (*p* < 0.05 vs. DF group; Figure 7B–D). Moreover, the protein expression of sodium-glucose cotransporter-2 (SGLT-2) in the kidneys was significantly increased in the DF group, but not in HF group, which could be effectively reversed by COS 1% and 5% supplementation (*p* < 0.05 vs. DF group; Figure 7E).

### 2.4. The Changes in Thiobarbituric Acid Reactive Substances (TBARS) Levels and Superoxide Dismutase (SOD) and Glutathione Peroxidase (GPx) Activities

As shown in Figure 8A, the levels of TBARS in the plasma and liver were significantly increased in DF group (*p* < 0.05 vs. NC group), but not in HF group, which could be effectively reversed by COS 1% and 5% supplementation (*p* < 0.05 vs. DF group). Supplementation of COS 0.5% significantly reversed the increased TBARS levels in the plasma (Figure 8Aa), but not in the livers (Figure 8Ab). Moreover, the liver SOD activity was decreased in DF group, but not HF group, compared to NC group (Figure 8Ba). The liver GPx activity was decreased in HF group, but not DF group, compared to NC group (Figure 8Bb). Supplementation of COS (0.5–5%) significantly increased the SOD activity in the liver of DF diabetic rats (*p* < 0.05 vs. HF or DF group; Figure 8Ba), but only COS 1% supplementation significantly increased the GPx activity (*p* < 0.05 vs. HF or DF group; Figure 8Bb).

## 3. Discussion

In the STZ diabetic animal models, the plasma level of insulin is declined remarkably after a single high dose of STZ administered with intraperitoneal injection that induces severe symptoms of diabetes, such as hyperglycemia, hyperphagia, polydipsia and polyuria; however, it has a high level of mortality rate [10]. The combination of HF diet and single dose of STZ administered with subcutaneous injection has been used as a diabetic animal model with lower mortality rate, which induced hyperglycemia and hepatic steatosis [8,9]. In this study, we found that the plasma levels of glucose, TC, and TG and the water intake and urine volume were markedly increased and the plasma insulin level was significantly decreased in the DF group compared to both NC and HF groups. We also found that the renal function markers BUN, creatine, and uric acid were significantly increased in the DF group compared to both NC and HF groups, indicating that the nephrotoxicity was induced in the DF diabetic rats. Supplementation of COS effectively reversed these diabetic symptoms in the DF diabetic rats. COS supplementation could also significantly reduce the increase in plasma TC, LDL-C + VLDL-C, and TG levels and AST activity in the DF diabetic rats; however, it could not significantly reverse the increased liver TG and TC levels and plasma ALT activity in the DF diabetic rats.

Liver plays an important role in maintaining plasma glucose concentration, which is regulated by the metabolic pathway of gluconeogenesis and glycolysis [11]. Gluconeogenesis is an anabolic pathway of glucose formation from non-hexose precursors, such as pyruvate, lactate, glycerol, and glucogenic amino acids. Gluconeogenesis is highly responsible for hepatic glucose production that is an essential mechanism for the maintenance of circulating blood glucose levels [12]. Glucose 6-phosphatase is one of the unique enzymes for gluconeogenesis that can dephosphorylate glucose 6-phosphate to form glucose, which is free to enter the blood circulation [12]. Hexokinase, one of the rate-limiting enzymes of glycolysis, can catalyze the phosphorylation of glucose by ATP to produce glucose-6-phosphate, which plays an important role in the maintenance of glucose homeostasis in cells [13]. Moreover, PEPCK is known to be one of the key enzymes for gluconeogenesis. The expression of PEPCK has been shown to be increased by p38 signaling activation in liver cells [14]. AMPK is an intracellular energy sensor. The 5-aminoimidazole-4-carboxamide riboside (AICAR), an AMPK activator, has been found to reduce the p38-activated PEPCK signaling-regulated gluconeogenesis in hepatocytes [15]. The upregulation of gluconeogenesis and glucose production in the liver is known to play an important role in diabetic hyperglycemia. The hepatic gluconeogenic signaling pathway has been suggested to be a therapeutic strategy for treating diabetes [14,16]. In the present study, the results showed that COS supplementation effectively enhanced the activity of hexokinase and reduced the glucose-6-phosphatase activity and increased the glycogen content in the livers of DF diabetic rats. Moreover, supplementation of COS markedly reduced the levels of liver PEPCK and phosphorylated p38 protein expression, but significantly enhanced the phosphorylation of liver AMPK protein. These results indicate that COS possesses the ability to reduce gluconeogenesis and enhance glycolysis in the livers, which may further contribute to the decrease in diabetic hyperglycemia.

Protein kinase C (PKC) is a serine/threonine kinase and an important component of the signal transduction system in organisms [17]. Diabetes causes oxidative stress, which further impairs insulin action [18]. The activation of PKC signaling pathway has also been found to increase oxidative stress and activate MAPK, contributing to diabetic nephropathy [19]. Katiyar et al. have reported that COS supplementation increases hepatic anti-oxidative enzyme activities including SOD and CAT, and decreased TBARS value in an alloxan-induced diabetic mouse model [20]. In the present study, the increased plasma and hepatic TBARS values and the decreased SOD and GPx activities in the DF diabetic rats could be significantly reversed by COS supplementation. Moreover, COS supplementation can reduce the protein expression levels of p-p38 MAPK and PKCα. These results suggest that COS supplementation ameliorates the lipid peroxidation and anti-oxidative enzymes activities in the DF diabetic rats, which may be through the reduction of PKCα and p38 MAPK signals.

The protein expression of SGLT2, a glucose transport protein, is markedly increased in the kidneys of diabetic patients that the glucose reabsorption from the renal proximal tubules is 20% more than that of normal subjects, causing the increase in blood glucose concentration, which can be effectively reversed by SGLT2 inhibitors, inhibiting the coupled reabsorption of sodium and glucose [21]. Here, we also found that supplementation of COS 1% and 5% significantly reduced the levels of protein expression of SGLT2, which may further reduce the plasma glucose concentration.

COS has recently been found to possess the beneficial effects on HF diet-induced non-alcoholic fatty liver disease in mice [22]. Recently, our previous study also found that COS supplementation could effectively reduce the levels of TC and TG in the plasma and liver of HF diet-induced obese rats [23]. Unexpectedly, the present study found that COS supplementation could not significantly reduce the increased levels of TC and TG in the liver of HF diet/STZ-induced diabetic rats. The catabolic and anabolic pathways for lipid and glucose metabolism in the liver are closely correlated and hardly be separated [24]. The involvement of the interaction of hepatic lipid and glucose metabolism during hyperglycemia and hepatic steatosis may affect the effects of COS on hepatic lipid accumulation. This issue needs to be clarified in the future.

COS has small molecular weight and higher absorption in the intestine, which results in permitting its quick access to the blood circulation [25]. COS has been shown to alleviate the impairment of blood glucose metabolism in a diabetic rat model [4]. Glucosamine is an ultimate degradation product of chitosan. Huang et al. have shown that both glucosamine and COSs (number-average molecular weight ≤ 1000 and ≤ 3000) can alleviate obesity and dyslipidemia by inhibiting the adipocyte differentiation in HFD-induced obese rats [26]. Hwang et al. have also found that glucosamine can improve body weight gain and insulin resistance in HFD-fed mice, although it increases body weight gain and reduces the hepatic insulin response in normal chow diet-fed mice [27]. The results of present study indicate that COS may activate the AMPK-regulated signaling cascade to reduce hepatic gluconeogenesis and enhance glycolysis, when circulating COS entered the liver, leading to the decrease in hyperglycemia. Moreover, when circulating COS entered the kidney, it may inhibit the renal SGLT2 expression, resulting in alleviating abnormal glucose metabolism.

## 4. Materials and Methods

### 4.1. Materials

COS was obtained from Koyo Chemical Co., Ltd. (Tokyo, Japan). The average molecular weight (MW) and deacetylation degree (DD) of COS were about 719 Da and 100%, respectively. The generally manufacturing process for COS in company was: 100% deacetylated chitosan -- > dissolve in hydrochloric acid -- > decompose with chitosanase -- > adjust pH value -- > inactivation -- > decolorization -- > dry -- > chitosan oligosaccharide. The kits for detection of glucose, uric acid, total cholesterol (TC), and triglyceride (TG) were purchased from Audit Diagnostics (Cork, Ireland). A rat insulin assay kit was obtained from Mercodia AB (Uppsala, Sweden). A creatinine enzymatic assay kit was provided by Eagle Diagnostics (Desoto, TX, USA). The assay kits for blood urea nitrogen (BUN), aspartate aminotransferase (AST), and alanine aminotransferase (ALT) were purchased from Randox (Antrium, UK). The superoxide dismutase (SOD) and glutathione peroxidase (GPx) assay kits were obtained from Cayman Chemical (Ann Arbor, MI, USA). A bicinchoninic acid (BCA) protein assay kit was purchased from Thermo Fisher Scientific (Waltham, MA, USA).

### 4.2. Animals

The animal experiments were approved by the Animal House Management Committee of the National Taiwan Ocean University and were conducted by the guidelines for care and use of laboratory animals. Five-week-old male Wistar rats were purchased from BioLASCO (Taipei, Taiwan). Animals were housed in cages in an animal room maintained at appropriate temperature and relative humidity and a 12 h light/12 h dark cycle. Animals could be free access to a standard rodent feed LabDiet Rodent 5001 (St. Louis, MO, USA) and water. Rats were divided into six groups: normal control (NC group), HF diet (HF group), STZ-induced diabetes with HF diet (DF group), DF with 0.5% COS (D0.5F group), DF with 1% COS (D1F group), and DF with 5% COS (D5F group). The dosages for COS were selected according to the previous reports [23,28] and our preliminary tests. These diet compositions were shown in Table 4. Rats received 50 mg/kg STZ in 0.1 M citrate by subcutaneous injection to induce diabetes. The experimental period was 4 weeks. Rats were weighed once a week and their food and water intake were weighed three times a week. Urine samples were collected during the last 3 days before animal euthanasia.

### 4.3. Sampling Blood and Tissue

Rats were euthanized with anesthesia after 4 weeks of experimental period. The samples of blood and tissues for liver, adipose, and kidney were collected. Plasma was prepared by centrifugation at 3000 rpm (1570× *g*) at 4 °C for 20 min. The plasma and tissue samples were stored in a freezer at −80 °C until further analysis.

### 4.4. Determination of Plasma Glucose, Insulin, Creatinine, BUN, Uric Acid, AST, and ALT

The levels of glucose, insulin, creatinine, BUN, and uric acid and the activities of AST, and ALT in the plasma samples were determined by specific kits or reagents as described in Materials Section 4.1.

### 4.5. Measurement of Liver SOD and GPx Activities and Plasma and Liver Lipid Peroxide (Thiobarbituric Acid Reactive Substances, TBARS) Contents

The liver SOD and GPx activities were determined by the SOD and GPx assay kits and the absorbance at 440 nm and 340 nm, respectively, were analyzed by a VersaMax microplate reader (Molecular Device, San Jose, CA, USA).

The measurement of TBARS was performed by the reaction between thiobarbituric acid (TBA) and lipid peroxide product (malondialdehyde, MDA) in the samples of plasma and liver lysate. The 1,1,3,3-tetraethoxypropane (Sigma-Aldrich, St. Louis, MO, USA) was as a standard group and the physiological saline was as a blank group. A Synergy HT microplate reader (BioTek, Winooski, VT, USA) was used to detect the fluorescence at Ex/Em 515/553 nm.

### 4.6. Measurement of Lipids in the Plasma, Liver, and Adipose Tissues and Plasma Lipoproteins

The levels of plasma TC and TG were measured using the enzymatic assay kits that the absorbance at 500 nm was examined by a spectrophotometer (UV/VIS-7800, JASCO International).

The extraction of lipids in the liver and adipose tissues was performed as previously described [29]. The TC and TG levels were determined by the TC and TG enzymatic assay kits (Audit Diagnostics).

According to the characteristics of different lipoprotein density, the ultracentrifugation method was used to separate and analyze the levels of lipoproteins in the plasma as previously described [30]. A Hitachi CP90NX ultracentrifuge with RPL42T rotor (Tokyo, Japan) was used to segregate the high-density lipoprotein cholesterol (HDL-C), low-density lipoprotein cholesterol (LDL-C), and very low-density lipoprotein cholesterol (VLDL-C) in the plasma by density gradient ultracentrifugation (194,000× *g* at 10 °C for 3 h). The HDL-C, LDL-C, and VLDL-C were then recovered by tube slicing.

### 4.7. Measurement of Glycogen Content and Activities of Hexokinase, Glucose-6-Phosphatase (G6Pase), and Glucose-6-Phosphate Dehydrogenase (G6PD)

The glycogen content was determined as previously described [31]. The liver homogenates in citrate were mixed with exo-1,4-α-glucosidase (EC 3.2.1.3) to liberate glucose. The glucose levels were detected by a glucose assay kit (Audit Diagnostics).

The activity of hexokinase was determined as previously described [32] with a modification. The samples and reagents (0.25 M Glycylglycine buffer (pH 7.5), 0.75 M MgSO_4_, 7.5 mM NADP, 0.75 M Glucose, 1 M KCl, liver cytosol preparations, 0.75 M ATP and 7 unit/mL G6PD) were mixed and added to a 96-well microplate. The absorbance at 340 nm was analyzed by a VersaMax microplate reader (Molecular Devices).

The detection of G6Pase activity was based on the inorganic phosphorus production. The samples and reagents (0.05 M Tris-maleate buffer (pH 7.5), 0.1 M glucose-6-phosphate, and liver cytosol preparations) were mixed at 37 °C for 40 min. This reaction could produce inorganic phosphorus. The inorganic phosphorus was determined by a microcolorimetric method as previously described [33]. 5% Trichloroacetic acid was added to the reaction mixture, and then supernatant was collected after centrifugation at 3000 rpm 4 °C for 30 min. The 5 N H_2_SO_4_, 2.5% ammonium molybdate, and 10% ascorbic acid were then added to finally react. The absorbance at 660 nm was detected by a VersaMax microplate reader (Molecular Devices).

G6PD activity was detected indirectly as readout of NADPH generation. The reaction mixture (0.05 M Tris-HCl buffer (pH 7.5), 0.2 M MgCl_2_, 0.025 M glucose-6-phosphate, 7.5 mM β-Nicotinamide denine dinucleotide phosphate, and liver cytosol preparations) was added to a 96-well microplate. The absorbance at 340 nm was analyzed by a VersaMax microplate reader (Molecular Devices).

### 4.8. Western Blot Analysis

The protein extraction and immunoblot were analyzed as previously described [34]. Briefly, the protein extraction of liver and kidney tissues was determined by radioimmunoprecipitation assay (RIPA) buffer with a cocktail of phosphatase and protease inhibitors (Thermo Fisher Scientific). The protein concentrations in the samples were measured by a BCA protein assay kit (Thermo Fisher Scientific). For Western blotting, proteins (30 μg) were separated by 8–12% SDS-PAGE gel, transferred to the polyvinylidene difluoride membranes (Bio-Rad, Hercules, CA, USA), and probed with primary antibodies specific for phosphorylated (p) p38 mitogen-activated protein kinase (MAPK) (sc-7973; 1:1000), p38 (sc-7972; 1:1000), phosphoenolpyruvate carboxykinase (PEPCK; sc-32879; 1:1000), GAPDH (sc-47724; 1:1000), β-actin (sc-47778; 1:1000) (Santa Cruz Biotechnology, Santa Cruz, CA, USA), phosphorylated (p) AMP-activated protein kinase α (p-AMPKα; #2531; 1:1000), AMPKα (#2532; 1:1000) (Cell Signaling Technology, Danvers, MA, USA), sodium-glucose cotransporter-2 (SGLT2; ab37296; 1:1000) (Abcam, Cambridge, MA, USA) and PKCα (610107; 1:1000) (BD Transduction Laboratories, San Jose, CA, USA). Each one of the membranes was subsequently probed with horseradish peroxidase-conjugated secondary antibodies (Cell Signaling Technology, Danvers, MA, USA). The expression of target protein was analyzed by an Enhanced Chemiluminescence kit (Bio-Rad) and exposed to a Fujifilm X-ray film (Tokyo, Japan). The protein bands were densitometrically quantified by Image J 1.8 software (National Institutes of Health, Bethesda, MD, USA).

### 4.9. Statistics

The values are presented as mean ± standard deviation (S.D.). The one-way analysis of variance (ANOVA) with a post-hoc test (Duncan’s multiple range test) was used to analyze the statistical difference performed by a software of IBM SPSS statistics (version: 22.0; Armonk, NY, USA). The evaluation was considered as a significant difference if the *p* value is ≤ 0.05. Values with different superscript letters (a, b, c) in the same row for each parameter (tables) or in columns (bar graph; figures) are significantly different (*p* < 0.05).

## 5. Conclusions

The findings of this study demonstrated the potential of plasma lipids-lowering and glucose-lowering effects of COS supplementation in a DF diabetic rat model. Supplementation of COS may possess a potential for alleviating abnormal glucose metabolism in diabetic rats through the inhibition in hepatic gluconeogenesis and lipid peroxidation and renal SGLT2 expression. Moreover, our results revealed that 0.5% of COS can effectively improve the abnormal lipid metabolism in diabetic rats, while both 1% and 5% of COS not only improve the abnormal glucose metabolism, but also the abnormal lipid metabolism in diabetic rats. Nevertheless, supplementation with COS in the DF diabetic rats could not significantly decrease the hepatic lipid accumulation. The previous findings suggested that 5% COS may have potential for toxicological side effect in a HF diet-fed condition [23]. Based on the above reasoning, the findings of present study suggest that 1% COS may be more suitable as a dose for improving diabetes.

## Figures and Tables

**Figure 1 marinedrugs-19-00360-f001:**
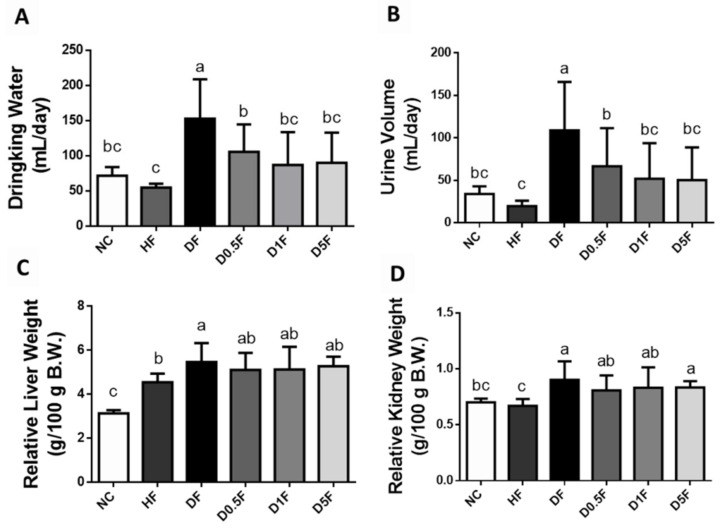
The changes in water intake (**A**), urine volume (**B**), and weights of liver (**C**) and kidney (**D**) in diabetic rats fed with different experiment diets for 4 weeks. Results are expressed as mean ± S.D. (*n* = 8) for each group. Values with different superscript letters (a, b, c) in columns are significantly different (*p* < 0.05) from one-way ANOVA followed by Duncan’s multiple range test. NC: Normal control diet. HF: High-fat (HF) diet. DF: STZ diabetes + HF diet. D0.5F: STZ diabetes + HF diet + 0.5% chitosan oligosaccharide (COS). D1F: STZ diabetes + HF diet + 1% COS. D5F: STZ diabetes + HF diet + 5% COS.

**Figure 2 marinedrugs-19-00360-f002:**
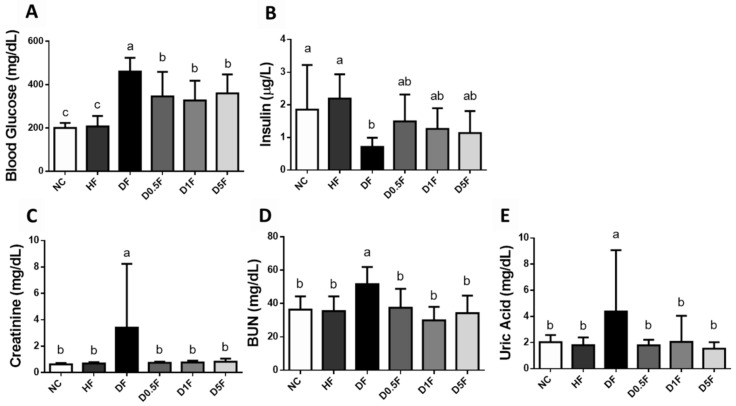
The changes in blood glucose (**A**), insulin (**B**), creatinine (**C**), BUN (**D**), and uric acid (**E**) in diabetic rats fed with different experiment diets for 4 weeks. Results are expressed as mean ± S.D. for each group (*n* = 7~8). Values with different superscript letters (a, b, c) in columns are significantly different (*p* < 0.05) from one-way ANOVA followed by Duncan’s multiple range test. NC: Normal control diet. HF: High-fat (HF) diet. DF: STZ diabetes + HF diet. D0.5F: STZ diabetes + HF diet + 0.5% chitosan oligosaccharide (COS). D1F: STZ diabetes + HF diet + 1% COS. D5F: STZ diabetes + HF diet + 5% COS.

**Figure 3 marinedrugs-19-00360-f003:**
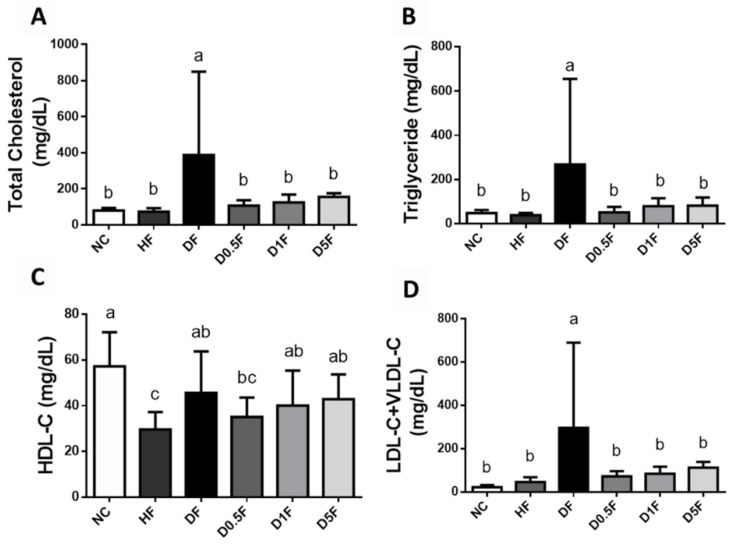
The changes in blood lipids in diabetic rats fed with different experiment diets for 4 weeks. The blood lipids of total cholesterol (**A**), triglyceride (**B**), HDL-C (**C**), and LDL-C + VLDL-C (**D**) were shown. Results are expressed as mean ± S.D. for each group (*n* = 7~8). Values with different superscript letters (a, b, c) in columns are significantly different (*p* < 0.05) from one-way ANOVA followed by Duncan’s multiple range test. NC: Normal control diet. HF: High-fat (HF) diet. DF: STZ diabetes + HF diet. D0.5F: STZ diabetes + HF diet + 0.5% chitosan oligosaccharide (COS). D1F: STZ diabetes + HF diet + 1% COS. D5F: STZ diabetes + HF diet + 5% COS.

**Figure 4 marinedrugs-19-00360-f004:**
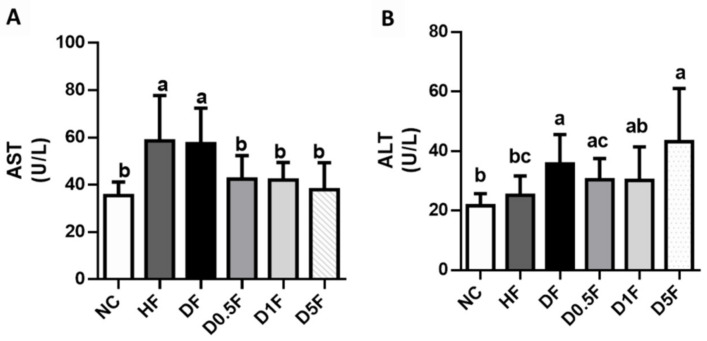
The changes in plasma AST (**A**) and ALT (**B**) activities in diabetic rats fed with different experiment diets for 4 weeks. Results are expressed as mean ± S.D. (*n* = 8) for each group. Values with different superscript letters (a, b, c) in columns are significantly different (*p* < 0.05) from one-way ANOVA followed by Duncan’s multiple range test. NC: Normal control diet. HF: High-fat (HF) diet. DF: STZ diabetes + HF diet. D0.5F: STZ diabetes + HF diet + 0.5% chitosan oligosaccharide (COS). D1F: STZ diabetes + HF diet + 1% COS. D5F: STZ diabetes + HF diet + 5% COS.

**Figure 5 marinedrugs-19-00360-f005:**
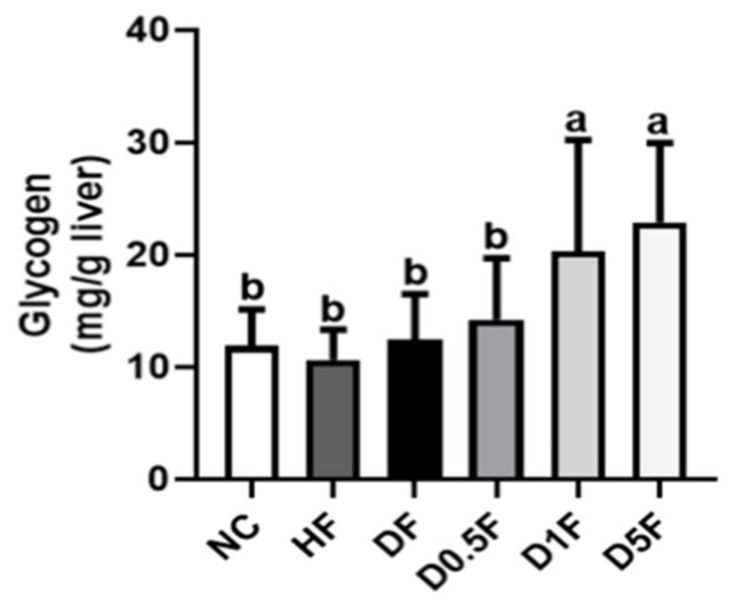
The changes in hepatic glycogen contents in diabetic rats fed with different experiment diets for 4 weeks. The glycogen (mg/g liver; A) and glycogen (g/liver; B) were shown. Results are expressed as mean ± S.D. (*n* = 8) for each group. Values with different superscript letters (a, b) in columns are significantly different (*p* < 0.05) from one-way ANOVA followed by Duncan’s multiple range test. NC: Normal control diet. HF: High-fat (HF) diet. DF: STZ diabetes + HF diet. D0.5F: STZ diabetes + HF diet + 0.5% chitosan oligosaccharide (COS). D1F: STZ diabetes + HF diet + 1% COS. D5F: STZ diabetes + HF diet + 5% COS.

**Figure 6 marinedrugs-19-00360-f006:**
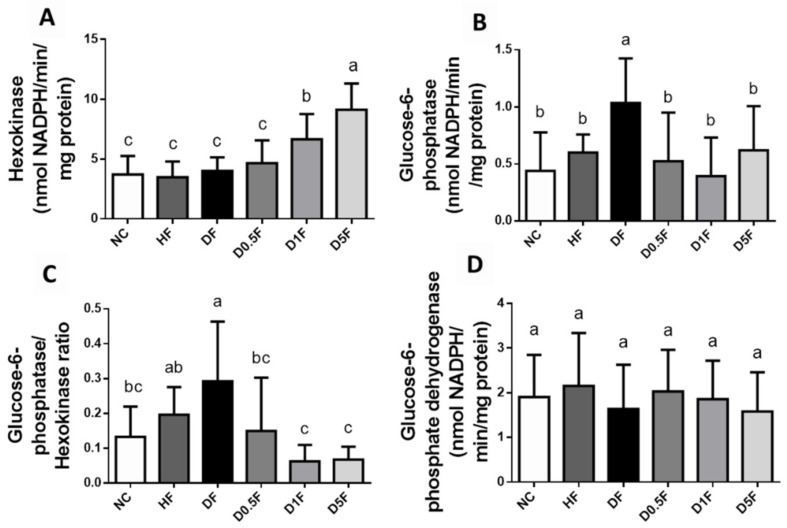
The changes in hepatic carbohydrate enzymes in diabetic rats fed with different experiment diets for 4 weeks. The activities of hexokinase (**A**), glucose-6-phosphatase (**B**), ratio of glucose-6-phosphatase/hexokinase (**C**), and glucose-6-phosphate dehydrogenase (**D**) were shown. Results are expressed as mean ± S.D. (*n* = 8) for each group. Values with different superscript letters (a, b, c) in columns are significantly different (*p* < 0.05) from one-way ANOVA followed by Duncan’s multiple range test. NC: Normal control diet. HF: High-fat (HF) diet. DF: STZ diabetes + HF diet. D0.5F: STZ diabetes + HF diet + 0.5% chitosan oligosaccharide (COS). D1F: STZ diabetes + HF diet + 1% COS. D5F: STZ diabetes + HF diet + 5% COS.

**Figure 7 marinedrugs-19-00360-f007:**
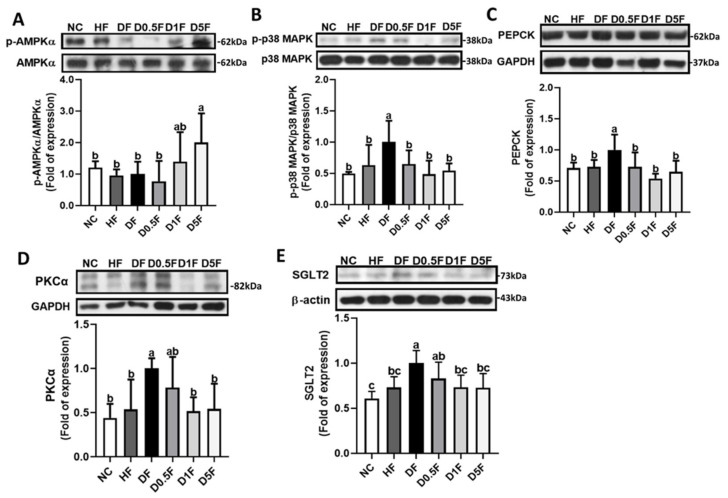
The changes in hepatic p-AMPKα/AMPKα (**A**), p-p38/p38 (**B**), PEPCK (**C**), and PKCα (**D**) and renal SGLT2 (**E**) protein expression in diabetic rats fed with different experimental diets for 4 weeks. Results are expressed as mean ± S.D. (*n* = 4–5) for each group. Values with different superscript letters (a, b, c) in columns are significantly different (*p* < 0.05) from one-way ANOVA followed by Duncan’s multiple range test. NC: Normal control diet. HF: High-fat (HF) diet. DF: STZ diabetes + HF diet. D0.5F: STZ diabetes + HF diet + 0.5% chitosan oligosaccharide (COS). D1F: STZ diabetes + HF diet + 1% COS. D5F: STZ diabetes + HF diet + 5% COS.

**Figure 8 marinedrugs-19-00360-f008:**
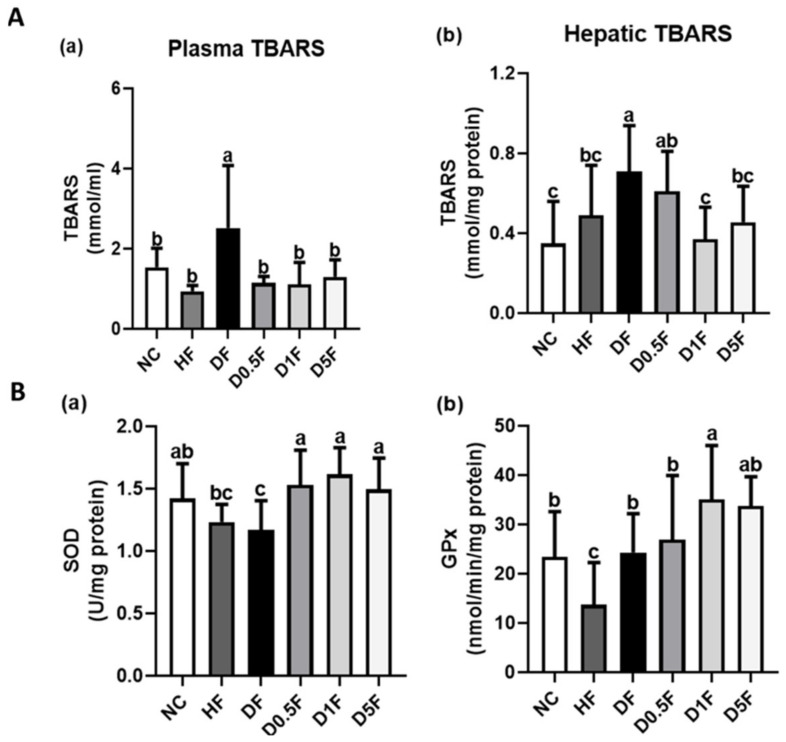
The changes in plasma (**Aa**) and hepatic (**Ab**) TBARS and hepatic SOD (**Ba**) and GPx (**Bb**) activities in diabetic rats fed with different experimental diets for 4 weeks. Results are expressed as mean ± S.D. (*n* = 8) for each group. Values with different superscript letters (a, b, c) in columns are significantly different (*p* < 0.05) from one-way ANOVA followed by Duncan’s multiple range test. NC: Normal control diet. HF: High-fat (HF) diet. DF: STZ diabetes + HF diet. D0.5F: STZ diabetes + HF diet + 0.5% chitosan oligosaccharide (COS). D1F: STZ diabetes + HF diet + 1% COS. D5F: STZ diabetes + HF diet + 5% COS.

**Table 1 marinedrugs-19-00360-t001:** The changes in body weight, food intake, drinking volume, urine volume, and weights of liver and kidney in diabetic rats fed with different experiment diets for 4 weeks.

Diet	NC	HF	DF	D0.5F	D1F	D5F
Initial body weight (g)	374.3 ± 23.7 ^a^	374.2 ± 19.6 ^a^	336.5 ± 31.8 ^b^	342.0 ± 16.4 ^b^	337.5 ± 30.7 ^b^	341.2 ± 22.8 ^b^
Final body weight (g)	484.3 ± 29.0 ^ab^	512.2 ± 22.2 ^a^	444.8 ± 51.9 ^b^	450.9 ± 26.0 ^b^	449.7 ± 38.1 ^b^	445.6 ± 44.3 ^b^
Body weight gain (g)	110.1 ± 14.9 ^b^	138.0 ± 17.7 ^a^	108.3 ± 29.2 ^b^	108.9 ± 21.0 ^b^	112.2 ± 25.7 ^b^	104.4 ± 25.4 ^b^
Food intake (g/day)	30.7 ± 2.12 ^ab^	28.4 ± 1.85 ^b^	35.1 ± 7.40 ^a^	33.4 ± 5.74 ^ab^	31.9 ± 5.38 ^ab^	32.1 ± 4.80 ^ab^
Feed efficiency (%) ^1^	3.58 ± 0.34 ^b^	4.84 ± 0.39 ^a^	3.15 ± 0.88 ^b^	3.35 ± 0.87 ^b^	3.67 ± 1.21 ^b^	3.38 ± 1.14 ^b^

Results are expressed as mean ± S.D. for each group (*n* = 8). Values with different superscript letters (a, b) in the same row for each parameter are significantly different (*p* < 0.05) from one-way ANOVA followed by Duncan’s multiple range test. NC: Normal control diet. HF: High fat diet. DF: Diabetes + high fat diet. D0.5F: Diabetes + high fat diet + 0.5% chitosan oligosaccharide. D1F: Diabetes + high fat diet + 1% chitosan oligosaccharide. D5F: Diabetes + high fat diet + 5% chitosan oligosaccharide. ^1^ Feed efficiency (%) = [body weight gain (g) ÷ food intake (g/day)] × 100%.

**Table 2 marinedrugs-19-00360-t002:** The changes in adipose tissue weights in diabetic rats fed with different experiment diets for 4 weeks.

Diet	NC	HF	DF	D0.5F	D1F	D5F
Total adipose tissue (g)	14.5 ± 3.52 ^a^	15.5 ± 2.46 ^a^	8.89 ± 4.39 ^b^	13.0 ± 4.64 ^a^	10.5 ± 2.64 ^ab^	10.1 ± 2.74 ^ab^
Relative adipose tissue weight (g/100g B.W.)	3.11 ± 0.66 ^a^	3.14 ± 0.54 ^a^	2.04 ± 0.83 ^b^	2.91 ± 0.96 ^a^	2.35 ± 0.41 ^ab^	2.33 ± 0.46 ^ab^
Perirenal adipose (g)	6.84 ± 1.82 ^a^	8.21 ± 1.30 ^a^	4.20 ± 2.88 ^b^	7.15 ± 2.77 ^a^	5.84 ± 1.72 ^ab^	5.05 ± 1.71 ^ab^
Relative perirenal adipose (g/100g B.W.)	1.48 ± 0.38 ^a^	1.67 ± 0.30 ^a^	0.98 ± 0.58 ^b^	1.59 ± 0.59 ^a^	1.30 ± 0.28 ^ab^	1.16 ± 0.30 ^a^^b^
Epididymal adipose (g)	6.60 ± 1.34 ^a^	7.83 ± 1.08 ^a^	4.81 ± 1.72 ^b^	5.95 ± 1.50 ^ab^	5.14 ± 0.70 ^ab^	5.23 ± 1.06 ^ab^
Relative epididymal adipose weight (g/100g B.W.)	1.45 ± 0.29 ^a^	1.59 ± 0.22 ^a^	1.08 ± 0.32 ^b^	1.34 ± 0.29 ^ab^	1.16 ± 0.08 ^b^	1.20 ± 0.19 ^ab^

Results are expressed as mean ± S.D. for each group (*n* = 8). Values with different superscript letters (a, b) in the same row for each parameter are significantly different (*p* < 0.05) from one-way ANOVA followed by Duncan’s multiple range test. NC: Normal control diets. HF: High-fat (HF) diets (lard 10%). DF: STZ diabetes + HF diet. D0.5F: STZ diabetes + HF diet + 0.5% chitosan oligosaccharide (COS). D1F: STZ diabetes + HF diet + 1% COS. D5F: STZ diabetes + HF diet + 5% COS.

**Table 3 marinedrugs-19-00360-t003:** The changes in hepatic lipids in diabetic rats fed with different experiment diets for 4 weeks.

Diet	NC	HF	DF	D0.5F	D1F	D5F
Triglyceride						
(mg/g liver)	13.8 ± 4.46 ^b^	43.6 ± 11.0 ^a^	47.9 ± 10.0 ^a^	48.3 ± 22.9 ^a^	41.9 ± 18.0 ^a^	34.0 ± 10.8 ^a^
(g/liver)	0.20 ± 0.06 ^b^	0.99 ± 0.31 ^a^	1.11 ± 0.30 ^a^	1.07 ± 0.50 ^a^	0.91 ± 0.34 ^a^	0.75 ± 0.22 ^a^
Total cholesterol						
(mg/g liver)	3.22 ± 0.49 ^c^	62.6 ± 8.50 ^b^	83.8 ± 12.5 ^a^	86.0 ± 17.4 ^a^	74.8 ± 10.9 ^a^	74.9 ± 10.7 ^a^
(g/liver)	0.05 ± 0.01 ^c^	1.41 ± 0.24 ^b^	1.94 ± 0.46 ^a^	1.97 ± 0.57 ^a^	1.67 ± 0.38 ^ab^	1.67 ± 0.26 ^ab^

Results are expressed as mean ± S.D. for each group (*n* = 8). Values with different superscript letters (a, b, c) in the same row for each parameter are significantly different (*p* < 0.05) from one-way ANOVA followed by Duncan’s multiple range test. NC: Normal control diet. HF: High fat diet. DF: Diabetes + high fat diet. D0.5F: Diabetes + high fat diet + 0.5% chitosan oligosaccharide. D1F: Diabetes + high fat diet + 1% chitosan oligosaccharide. D5F: Diabetes + high fat diet + 5% chitosan oligosaccharide.

**Table 4 marinedrugs-19-00360-t004:** Composition of experimental diets (%).

Ingredient (%)	NC	HF	DF	D0.5F	D1F	D5F
Chow diet	100	89.4	89.4	88.9	88.4	84.4
Lard		10	10	10	10	10
Cholesterol		0.5	0.5	0.5	0.5	0.5
Cholic acid		0.1	0.1	0.1	0.1	0.1
Chitosan oligosaccharide ^1^				0.5	1	5
Total calories (kcal/100g)	336.2	390.6	390.6	389.9	389.2	383.8
Carbohydrates (% kcal)	57.9	44.6	44.6	44.6	44.7	45.42
Protein (% kcal)	28.7	22.1	22.1	22.0	21.9	21.22
Fat (% kcal)	13.4	33.3	33.3	33.4	33.4	33.36
	100	100	100	100	100	100

^1^ The average MW and DD of chitosan oligosaccharide are about 719 Dalton and 100%, respectively. NC: Normal control diets; HF: High fat diets (Lard 10%); DF: Diabetes + high fat diet; D0.5F: Diabetes + high fat diet + 0.5% chitosan oligosaccharide; D1F: Diabetes + high fat diet + 1% chitosan oligosaccharide; D5F: Diabetes + high fat diet + 5% chitosan oligosaccharide.

## Data Availability

The data presented in this study are available from the corresponding author upon reasonable request.

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
