# Peer review of "Chitosan Oligosaccharide Alleviates Abnormal Glucose Metabolism without Inhibition of Hepatic Lipid Accumulation in a High-Fat Diet/Streptozotocin-Induced Diabetic Rat Model"

_marinedrugs, 2021, doi:10.3390/md19070360_

Round 1

Reviewer 1 Report

The authors showed the effects of COS on STZ-induced diabetic rat model such as lowering of plasma glucose and plasma lipids. This article submitted by Liu et al is of relevance in the field, well written and interesting to read. However, some issues that have to be addressed:

  1. Please describe the rationale for the COS dosage in vivo experiment.
  2. The authors should describe how COS is manufactured in company.
  3. The authors should discuss how non-digestible COS alleviates glucose metabolism in vivo systems. 

Author Response

Reviewer 1:       

The authors showed the effects of COS on STZ-induced diabetic rat model such as lowering of plasma glucose and plasma lipids. This article submitted by Liu et al is of relevance in the field, well written and interesting to read. However, some issues that have to be addressed:

  1. Please describe the rationale for the COS dosage in vivo experiment.

Response: We appreciate the reviewer's comment. We have added a description for the rationale for the COS dosage in vivo experiment in the Methods section of this revised manuscript according to the suggestion of reviewer.

The dosages for COS were selected according to the previous reports [23,25] and our preliminary tests.

  1. The authors should describe how COS is manufactured in company.

Response: We appreciate the reviewer's comment. We have added a description for the generally manufacturing process for COS in company in the Methods section of this revised manuscript according to the suggestion of reviewer.

The generally manufacturing process for COS in company was: 100% deacetylated chitosan --> dissolve in hydrochloric acid --> decompose with chitosanase --> adjust pH value --> inactivation --> decolorization --> dry --> chitosan oligosaccharide.

  1. The authors should discuss how non-digestible COS alleviates glucose metabolism in vivo systems. 

Response: We appreciate the reviewer's comment. We have discussed this issue in the Discussion section of this revised manuscript according to the suggestion of reviewer.

COS has small molecular weight and higher absorption in the intestine, which re-sults in permitting its quick access to the blood circulation [25]. COS has been shown to alleviate the impairment of blood glucose metabolism in a diabetic rat model [4]. Glucosamine is an ultimate degradation product of chitosan. Huang et al. have shown that both glucosamine and COSs (number-average molecular weight ≤1000 and ≤3000) can alleviate obesity and dyslipidemia by inhibiting the adipocyte differentiation in HFD-induced obese rats [26]. Hwang et al. have also found that glucosamine can im-prove body weight gain and insulin resistance in HFD-fed mice, although it increases body weight gain and reduces the hepatic insulin response in normal chow diet-fed mice [27]. The results of present study indicate that COS may activate the AMPK-regulated signaling cascade to reduce hepatic gluconeogenesis and enhance glycolysis, when circulating COS entered the liver, leading to the decrease in hyperglycemia. Moreover, when circulating COS entered the kidney, it may inhibit the renal SGLT2 expression, resulting in alleviating abnormal glucose metabolism.

Reviewer 2 Report

In the paper named “Chitosan oligosaccharide alleviates abnormal glucose metabo-2 lism without inhibition of hepatic lipid accumulation in a high-fat diet/streptozotocin-induced diabetic rat model” authors investigate the effects of chitosan oligosaccharide (COS) on glucose metabolism and hepatic steatosis in a high-fat (HF) diet/streptozotocin-induced diabetic rat model.

The results of this work, suggest that COS may possess a potential for alleviating abnormal glucose metabolism in diabetic rats through the inhibition of hepatic gluconenogenesis and lipid peroxidation and renal SGLT2 expression.

However there are some points that should be clarified by the authors

The tables are difficult to follow due to the amount of data and the meanings are also complicated. As a suggestion to the authors, I would try to complement the content of the tables with some graphs that show many of the results in a simpler way with bars.

  • Throughout the work there are many abbreviations that have not been mentioned previously, and that should have their meaning to make it easier to read
  • Although the authors carry out the experiments with the HF group and the data for this group are in all the tables, it is not referred to, except in the discussion. In some of the cases the results shown by this group are interesting.
  • The tables are difficult to follow due to the amount of data and the meanings are also complicated. As a suggestion to the authors, I would try to complement the tables with some graphs that show many of the results in a simpler way with bars.
  • In the table 4 captions there was lost part of the explication.
  • In figure 1, 2 and 3 captions there was lost the parts A, B,C.. explication
  • In point 2.3 between lines 133 to 136 is difficult The explanation is difficult to follow because part of the data is in figure 2 and part in table 4, but it would be necessary to specify it more clearly. for example putting next to hexokinase activity on line 135 (table4)
  • Why author present the some data as mg/g liver and g/liver??
  • In point 2.4 what means (fig 4A-a), Fig 4A-b.... this a and b are referred to the significance??

Author Response

Reviewer 2:

In the paper named “Chitosan oligosaccharide alleviates abnormal glucose metabo-2 lism without inhibition of hepatic lipid accumulation in a high-fat diet/streptozotocin-induced diabetic rat model” authors investigate the effects of chitosan oligosaccharide (COS) on glucose metabolism and hepatic steatosis in a high-fat (HF) diet/streptozotocin-induced diabetic rat model. The results of this work, suggest that COS may possess a potential for alleviating abnormal glucose metabolism in diabetic rats through the inhibition of hepatic gluconenogenesis and lipid peroxidation and renal SGLT2 expression.

However, there are some points that should be clarified by the authors

  1. The tables are difficult to follow due to the amount of data and the meanings are also complicated. As a suggestion to the authors, I would try to complement the content of the tables with some graphs that show many of the results in a simpler way with bars. Throughout the work there are many abbreviations that have not been mentioned previously, and that should have their meaning to make it easier to read. Although the authors carry out the experiments with the HF group and the data for this group are in all the tables, it is not referred to, except in the discussion. In some of the cases the results shown by this group are interesting. The tables are difficult to follow due to the amount of data and the meanings are also complicated. As a suggestion to the authors, I would try to complement the tables with some graphs that show many of the results in a simpler way with bars.

Response: We appreciate the reviewer's comment. We have revised the manuscript according to the suggestion of reviewer. The abbreviations have been mentioned previously. We revised the descriptions for the data presentation in the Results section. Many data from Tables were transferred to the bar graphs (additional 4 figures-Figs. 1, 2, 3, and 6 of this revised manuscript).

  1. In the table 4 captions there was lost part of the explication.

Response: We appreciate the reviewer's comment. We have revised the Table 4 according to the suggestion of reviewer. We also transferred some data from Table 4 to Figure 6.

  1. In figure 1, 2 and 3 captions there was lost the parts A, B,C.. explication

Response: We appreciate the reviewer's comment. We have revised the captions of these figures according to the suggestion of reviewer.

  1. In point 2.3 between lines 133 to 136 is difficult The explanation is difficult to follow because part of the data is in figure 2 and part in table 4, but it would be necessary to specify it more clearly. for example putting next to hexokinase activity on line 135 (table4)

Response: We appreciate the reviewer's comment. We have revised the Table 4 and the descriptions of results according to the suggestion of reviewer. We have re-organized the original Table 4 that the data of hepatic carbohydrate enzymes are transferred to Figure 6.

  1. Why author present the some data as mg/g liver and g/liver??

Response: We appreciate the reviewer's comment. We revised this figure according to the suggestion of reviewer. There are two different representation methods for glycogen; however, to avoid confusion, we removed a figure for g/liver and kept a figure for mg/g liver.

  1. In point 2.4 what means (fig 4A-a), Fig 4A-b.... this a and b are referred to the significance??

Response: We appreciate the reviewer's comment. We have added the descriptions in the caption of this figure (Figure 8 of this revised manuscript; original figure 4). The (a) and (b) are sub-figures that the changes in plasma (A-a) and hepatic (A-b) TBARS and hepatic SOD (B-a) and GPx (B-b) activities were shown.

Reviewer 3 Report

The present study aims to investigate the effects and possible mechanisms of chitosan oligosaccharides as a dietary supplement on abnormal glucose metabolism in a diabetic rat model fed with high-fat diet, which induced hyperglycemia and hepatic steatosis. The authors conclude that chitosan oligosaccharides may possess a potential for alleviating abnormal glucose metabolism in diabetic rats through the inhibition of hepatic gluconeogenesis and lipid peroxidation and renal SGLT2 expression. This study follows previous studies of the same group regarding the effect of chitosan oligosaccharides as a dietary supplement on hepatic lipogenesis, lipolysis, etc. The work is interesting, however there are several points, requiring correction and/or clarification to be the manuscript acceptable for publication.

Major points

  1. The presentation of results needs better organization to help the readers to understand the findings of the study. It is not clear what the subscript letters in the various Tables are showing. For example, it is noted in Table 3 that “Values with different superscript letters (a, b, c) mean significantly with each other (p < 0.05)”. So, what it is shown by superscript letters “a,b” in insulin analysis (second line) of NC animals? Similar questions are raised in all Tables and Figures. The authors should explain the comparisons made either in section 4.9 or in each one of the Tables/Figures. Moreover, the comparison in Tables/Figures should be in line with the respective in the text. For example, it is written in text (lines 110-112) “Supplementation of COS (0.5-5%) significantly reversed the change in plasma AST activity in the DF diabetic rats (p<0.05 vs DF group; Fig. 1)”, however in the respective figure the statistical significance to HF rats is shown. The authors should carefully proceed with the comparisons needed and carefully note the groups subjected to comparison.
  2. Line 44: Glucosamine is not a polysaccharide. Revise appropriately.
  3. Line 85: correct the number of the Table.
  4. Line 176: the heading 2.4 is exactly the same as the previous (2.3; line 131).
  5. Line 268: According the average MW written, the mean size of chitosan oligosaccharides used is that of tetrasaccharides. Therefore, the authors should discuss their findings with the findings observed when glucosamine was used as supplement in high-fat diet.

Minor points

  1. Lines 33-34: correct grammar errors.
  2. Line 41: Chitin is a polysaccharide not generally found in plants and animals; only selected organisms produce chitin.
  3. Lines 45-46: revise appropriately.
  4. Lines 269-277: Most of the materials described are also described in sections 4.4, 4.5, 4.6 and 4.8.
  5. Line 295: replace “these samples” with “plasma and tissue samples”.
  6. Line 358: delete the word “buffer”.
  7. Line 361: use more appropriate word instead of “segregated”.
  8. Line 369: replace “the membrane” with “each one of the membranes”.

Author Response

Reviewer 3:

The present study aims to investigate the effects and possible mechanisms of chitosan oligosaccharides as a dietary supplement on abnormal glucose metabolism in a diabetic rat model fed with high-fat diet, which induced hyperglycemia and hepatic steatosis. The authors conclude that chitosan oligosaccharides may possess a potential for alleviating abnormal glucose metabolism in diabetic rats through the inhibition of hepatic gluconeogenesis and lipid peroxidation and renal SGLT2 expression. This study follows previous studies of the same group regarding the effect of chitosan oligosaccharides as a dietary supplement on hepatic lipogenesis, lipolysis, etc. The work is interesting, however there are several points, requiring correction and/or clarification to be the manuscript acceptable for publication.

Major points

  1. The presentation of results needs better organization to help the readers to understand the findings of the study. It is not clear what the subscript letters in the various Tables are showing. For example, it is noted in Table 3 that “Values with different superscript letters (a, b, c) mean significantly with each other (p < 0.05)”. So, what it is shown by superscript letters “a,b” in insulin analysis (second line) of NC animals? Similar questions are raised in all Tables and Figures. The authors should explain the comparisons made either in section 4.9 or in each one of the Tables/Figures. Moreover, the comparison in Tables/Figures should be in line with the respective in the text. For example, it is written in text (lines 110-112) “Supplementation of COS (0.5-5%) significantly reversed the change in plasma AST activity in the DF diabetic rats (p<0.05 vs DF group; Fig. 1)”, however in the respective figure the statistical significance to HF rats is shown. The authors should carefully proceed with the comparisons needed and carefully note the groups subjected to comparison.

Response: We appreciate the reviewer's comment. We have carefully revised this manuscript according to the suggestion of reviewer.

We revised the descriptions for the data presentation in the Results section of this revised manuscript. Moreover, many data from Tables were transferred to the bar graphs (the additional 4 figures - Figs. 1, 2, 3, and 6 were added) according to the suggestion of reviewer 2. We also revised the descriptions for statistical significance. The one-way analysis of variance (ANOVA) with a post-hoc test (Duncan’s multiple range test) was used to analyze the statistical difference. Values with different superscript letters (a, b, c) in the same row for each parameter (tables) or in columns (bar graph; figures) are significantly different (p < 0.05). (For example: for Insulin levels: NC group- a, HF group- a, DF group- b, D0.5F group- a,b: no significant difference between NC and HF or between DF and D0.5F or among NC, HF, and D0.5F; significant difference between NC and DF or between HF and DF).

  1. Line 44: Glucosamine is not a polysaccharide. Revise appropriately.

Response: We appreciate the reviewer's comment. We have revised the description for this issue in the Introduction section of this revised manuscript according to the suggestion of reviewer.

  1. Line 85: correct the number of the Table.

Response: We appreciate the reviewer's comment. We have corrected this issue in this revised manuscript. Data of original Table 3 have been transferred to Figure 2 of this revised manuscript (suggested by reviewer 2).

  1. Line 176: the heading 2.4 is exactly the same as the previous (2.3; line 131).

Response: We appreciate the reviewer's comment. We have corrected this issue in this revised manuscript.

  1. Line 268: According the average MW written, the mean size of chitosan oligosaccharides used is that of tetrasaccharides. Therefore, the authors should discuss their findings with the findings observed when glucosamine was used as supplement in high-fat diet.

Response: We appreciate the reviewer's comment. We have discussed this issue in the Discussion section of this revised manuscript according to the suggestion of reviewer.

Minor points

  1. Lines 33-34: correct grammar errors.

Response: We appreciate the reviewer's comment. We have corrected this issue in this revised manuscript.

  1. Line 41: Chitin is a polysaccharide not generally found in plants and animals; only selected organisms produce chitin.

Response: We appreciate the reviewer's comment. We have corrected this issue in this revised manuscript.

  1. Lines 45-46: revise appropriately.

Response: We appreciate the reviewer's comment. We have revised this sentence in this revised manuscript.

  1. Lines 269-277: Most of the materials described are also described in sections 4.4, 4.5, 4.6 and 4.8.

Response: We appreciate the reviewer's comment. We have revised these sentences in this revised manuscript.

  1. Line 295: replace “these samples” with “plasma and tissue samples”.

Line 358: delete the word “buffer”.

Response: We appreciate the reviewer's comment. We have revised these words in this revised manuscript.

  1. Line 361: use more appropriate word instead of “segregated”.

Response: We appreciate the reviewer's comment. We have revised this word in this revised manuscript.

  1. Line 369: replace “the membrane” with “each one of the membranes”.

Response: We appreciate the reviewer's comment. We have revised these words in this revised manuscript.

Round 2

Reviewer 3 Report

The manuscript is a review version of a previously submitted work. The authors have revised the text following most of the reviewers’ comments. The manuscript can be now accepted for publication.